# Strongly ROS-Correlated, Time-Dependent, and Selective Antiproliferative Effects of Synthesized Nano Vesicles on BRAF Mutant Melanoma Cells and Their Hyaluronic Acid-Based Hydrogel Formulation

**DOI:** 10.3390/ijms251810071

**Published:** 2024-09-19

**Authors:** Silvana Alfei, Guendalina Zuccari, Constantinos M. Athanassopoulos, Cinzia Domenicotti, Barbara Marengo

**Affiliations:** 1Department of Pharmacy, University of Genoa, Viale Cembrano, 16148 Genoa, Italy; guendalina.zuccari@unige.it; 2Laboratory of Experimental Therapies in Oncology, IRCCS Istituto Giannina Gaslini, Via G. Gaslini 5, 16147 Genoa, Italy; 3Department of Chemistry, University of Patras, University Campus Rio Achaias, 26504 Patras, Greece; kath@chemistry.upatras.gr; 4Department of Experimental Medicine (DIMES), University of Genova, Via Alberti L.B., 16132 Genoa, Italy; cinzia.domenicotti@unige.it; 5IRCCS Ospedale Policlinico San Martino, 16132 Genoa, Italy

**Keywords:** cutaneous metastatic melanoma (CMM), triphenyl phosphonium (TPP) groups, mitochondria-targeting molecules, nanosized bola amphiphiles vesicles, cytotoxicity studies, low hemolytic effects, hydrogel formulation, biodegradability, high porosity, high swelling

## Abstract

Cutaneous metastatic melanoma (CMM) is the most aggressive form of skin cancer with a poor prognosis. Drug-induced secondary tumorigenesis and the emergency of drug resistance worsen an already worrying scenario, thus rendering urgent the development of new treatments not dealing with mutable cellular processes. Triphenyl phosphonium salts (TPPSs), in addiction to acting as cytoplasmic membrane disruptors, are reported to be mitochondria-targeting compounds, exerting anticancer effects mainly by damaging their membranes and causing depolarization, impairing mitochondria functions and their DNA, triggering oxidative stress (OS), and priming primarily apoptotic cell death. TPP-based bola amphiphiles are capable of self-forming nanoparticles (NPs) with enhanced biological properties, as commonly observed for nanomaterials. Already employed in several other biomedical applications, the per se selective potent antibacterial effects of a TPP bola amphiphile have only recently been demonstrated on 50 multidrug resistant (MDR) clinical superbugs, as well as its exceptional and selective anticancer properties on sensitive and MDR neuroblastoma cells. Here, aiming at finding new molecules possibly developable as new treatments for counteracting CMM, the effects of this TPP-based bola amphiphile (BPPB) have been investigated against two BRAF mutants CMM cell lines (MeOV and MeTRAV) with excellent results (even IC_50_ = 49 nM on MeOV after 72 h treatment). With these findings and considering the low cytotoxicity of BPPB against different mammalian non-tumoral cell lines and red blood cells (RBCs, selectivity indexes up to 299 on MeOV after 72 h treatment), the possible future development of BPPB as topical treatment for CMM lesions was presumed. With this aim, a biodegradable hyaluronic acid (HA)-based hydrogel formulation (HA-BPPB-HG) was prepared without using any potentially toxic crosslinking agents simply by dispersing suitable amounts of the two ingredients in water and sonicating under gentle heating. HA-BPPB-HA was completely characterized, with promising outcomes such as high swelling capability, high porosity, and viscous elastic rheological behavior.

## 1. Introduction

### 1.1. Melanoma

Malignant cells intended as metastatic melanomas (MMs) represent a historically formidable challenge in oncology, particularly when in advanced stages [1]. MMs are considered the most aggressive form of skin cancer, whose treatment options, thanks to incessant research, have undergone a remarkable transformation in the past decade, mainly due to the advent of immune checkpoint inhibitors (ICIs) and targeted therapies [2,3]. Standard treatment options include surgery, radiation therapy, systemic chemotherapy, targeted therapy, and immunotherapy, as well as their combinations [4]. MM is a malignancy with poor prognosis, encompassing diverse subtypes, each characterized by distinct clinical presentations, as well as histopathological and genetic features, which need specific treatments. The recent findings have evidenced the heterogeneity of melanomas, thus prompting a paradigm shift in melanomas classification and management [5]. The most prevalent form of melanoma is cutaneous melanoma (CM), caused by sun rays in most cases, in turn divided into subtypes based on sun exposure time, including chronic sun-damaged melanomas (CSDMs) and non-CSDMs. Acral melanomas (AMs) are a more particular form of CM [6,7,8]. Based on the degree of sun damage, superficial spreading cutaneous melanomas (SSMs), or lentigo malignant melanomas (LMMs) can also arise [5]. SSMs are predominantly observed in younger adults and are characterized by BRAF^V600E^ mutations, while LMMs feature alterations in NF1 or NRAS genes and can arise in sun-damaged skin of older individuals [8]. Desmoplastic melanoma (DM) is a rare subtype of melanoma, difficult to diagnose, commonly found in sun-damaged skin of the elderly [6]. Acral melanoma (AM) is a sporadic subtype of melanoma, not depending on sun exposure, fair skin type, family history of melanoma, or pre-existing melanocytic hyper pigmentated lesions [9,10]. AM arises on palms of the hands, soles of the feet, and in the subungual regions (nail beds) [9,10,11]. AM shows a lower tumor mutational burden (TMB), a different pattern of oncogenic driver mutation expression, a unique genomic aberration, an altered tumor environment, and less immune infiltrate compared to cutaneous melanoma [6,12,13,14]. Although it has been suggested that mechanical stress could be a causative factor for AM, the evidence for this assumption remains conflicting [15]. Less common but more complex forms of melanoma are mucosal and uveal melanomas (MUMs and UMs). MUMs are not caused by UV-dependent mutations, but by more complex genomic rearrangements [6]. Arising from melanocytes in mucosal tissues, MUM represents a challenge due to its occult locations and aggressive nature. In contrast, UMs, originating from melanocytes within the vascular layer of the eye, are instead characterized by GNAQ and GNA11 mutations and lack the characteristic BRAF mutations commonly detected in CMs [6]. The identified genetic aberrations associated to the specific subtypes of melanoma have allowed and are allowing researchers to develop novel and more effective therapeutic strategies and opportunely devised treatment plans, capable to cure rare melanoma forms which require specialized management strategies [1].

Concerning clinically approved drugs to treat CM, vemurafenib, dabrafenib, and encorafenib have proved to selectively inhibit activated BRAF^V600E/K/R/D^ monomers (class I BRAF mutants) in CM cancer cells, with an initial promising clinical success and wide therapeutic index [16].

Generally, the chemotherapy of malignant neoplasms, including several forms of MMs, often leads to drug-induced secondary tumorigenesis, due to nuclear DNA damage caused by traditional genotoxic chemotherapeutic agents. Furthermore, such diseases commonly are worsened by the emergency of drug resistance, due to adaptive genetic mechanisms of cancer cells, including drug inactivation, altered drug targets, adaptive responses, and dysfunctional apoptosis [17].

The use of a combinatorial treatment approach could delay the emergence of resistance and improve patient outcomes. Additionally, undesired drug–drug interactions could negatively affect the therapy outcomes [18].

Recently, oxidative therapies (OTs), demonstrating low tendency to develop resistance [19] and working by means of reactive oxygen species (ROS) induction by different methods [19,20], have been reviewed and reported as effective and alternative extra-genomic treatments for curing severe skin diseases and infections caused by MDR pathogens and by biofilm-producing microorganisms [21]. Similarly, high levels of oxidative stress (OS) caused by ROS induction have been reported as a novel type of anticancer therapy, capable of modulating the antitumor immune response without prompting resistance [22]. Specifically, photodynamic therapy (PDT) represents a promising method of tumor ablation and function-preserving oncological intervention, based on ROS induction by proper light-activated photosensitizers in presence of oxygen [19]. It is minimally invasive, repeatable, and has low side effects and no cumulative toxicity [23]. The combination of PDT with immune response stimulation might be the key to overcome CMM resistance and obtain better, sustainable clinical results [23].

Topical treatments involving ROS induced by PDT as antitumor agents are already clinically approved and have been widely employed against various tumors to which irradiation can be applied directly, such as lung, esophageal, gastric, breast, head and neck, bladder, and prostate carcinomas [24]. Some unavoidable disadvantages, including limited light penetration depth, poor tumor selectivity, and oxygen dependence, largely limit PDT therapeutic efficiency for solid tumors treatment of MM, as the objective of the present study [24]. Moreover, further in-depth clinical investigations are mandatory on the possible toxic effects of extensive ROS administration to mammalian cells, as well as on biosafety and target specificity [25,26].

Therefore, the search for synthetic compounds, with a mechanism of antitumor action not involving mutable pathways in cancer CMM cells and not contrastable by adaptative responses, is of paramount importance to avid possible recurrence in CMs.

Thus, the discovery of new anticancer molecules acting via mechanisms not involving cancer cells vital processes and which cells can transmute by genomic mutations remains urgent.

### 1.2. Mitochondria-Targeting Compounds

Mitochondria are the most promising target organelles for effective anticancer therapies, due to their key role in energy production, apoptosis induction, and ROS generation [27]. They possess their own DNA, independent and not associated with the mutable genetic material of cancer cells. Therefore, opportunely structured, cationic mitochondria-targeting compounds, able to interact and easily cross their strongly negative membranes and capable of a 100–1000 times accumulation inside them, can impair their functions, with irreversible damage for cells, bypassing the genetic mechanisms underlying tumor recurrence and resistance, and thus also being active in resistive cells. Particularly, molecules bearing the triphenyl phosphonium (TPP) moiety are the most widely used mitochondria-targeting carriers [28,29]. Mono- and, to a major extent, *bis*-TPP salts, thanks to the moderate cationic nature of their headgroup(s) due to a prolonged delocalization of the phosphorous atoms positive charge on the phenyl rings, maintain a high ability in interacting and crossing the negatively hyper-polarized cell membranes, such as those of bacteria and tumors [29,30,31,32], while exerting a limited cytotoxicity [29,32]. Targeting the cancer cytoplasmic membrane, TPP salts damage the phospholipid bilayer, thus triggering detrimental results for the cell, while facilitating their ingress [29]. Once inside the cancer cell, TPP-based compounds can target mitochondria, interact, and even cross their two highly negatively charged membranes, causing depolarization and/or destruction [29,33]. The consequent inhibitory activity on the mitochondria functions determines mitochondrial toxicity, permanent damage, oxidative stress (OS), and programmed death of cancer cells, by apoptosis, necroptosis, or autophagy [17,29,33]. On the contrary, due to the strong difference in the negative electric membrane potentials of both cytoplasmic and mitochondrial membranes between tumor and normal cells, these compounds may selectively accumulate in the mitochondria of cancer cells, rather than in those of normal cells [34], thus resulting in low levels of cytotoxicity [30]. Among *bis* cationic compounds, bola amphiphiles (BAs) are a particular class of cationic surfactants featuring one or more hydrophobic chains connecting two identical or different hydrophilic headgroups, endowed with nonpareil colloidal properties [29]. They exhibit unique hierarchically self-assembled structures, characterized by high polymorphism, thus being attractive for applications in various fields, including drug delivery, gene delivery, electronics, medical imaging, etc. [35]. *Bis*-triphenyl phosphonium (BTPP)-based bola amphiphiles (BTPP-BAs) have been reported to be mitochondria-target compounds, due to the TPP moiety, with high ability in crossing the mitochondrial membranes and accumulating inside them [36]. BTPP-BAs, featuring chains of 12, 16, 20, and 30 methylene units, have evidenced the capability to form nanosized vesicles and larger aggregates in water solutions, depending on the solutions’ concentrations [36]. Additionally, it seems that the colloidal properties of BTPP-BAs may play a central role in their mitochondrial toxicity [36], thus making possible enhanced antitumor effects compared to traditional TPP salts. Both alkyl triphenyl phosphonium salts [31,37,38,39] and pegylated *bis*-triphenyl phosphonium salts [40] have been successfully tested as antibacterial agents. Recently, BTPP-BA nano vesicles of 49 nm have been reported to have high potential to be developed as new curative options to treat severe human infections sustained by intractable pathogens, due to their selectivity for bacteria and low toxicity on eukaryotic cells [32]. While traditional alkyl TPP salts have also been extensively studied as anticancer molecules [41,42,43,44], to our knowledge, only recently has a study on the potent and selective anticancer effects of a BTPP-BA nanomaterial against sensitive (HTLA-230) and MDR (HTLA-ER) neuroblastoma (NB) cells been published [29]. Before this recent study, BAs have been extensively studied for other several applications, but not as anticancer agents per se [35].

### 1.3. CMM Topical Treatments: Hydrogel Formulations

The increasing emergence of drug resistance, high toxicity, and systemic side effect of chemotherapy in the treatment of cancers, including CMM, are the biggest daily challenges for the experts in the field [45,46,47]. In this regard, the local therapy strategy for treating skin cancer such as CMM can have great potential because of its advantages over other administration modes [48]. By topical administration, properly structured drug delivery systems can have the capacity of distributing high concentrations of the pharmaceutical active compounds just on tumor sites, reducing side effects [48]. Among different drug delivery systems, hydrogels constitute ideal platforms for local therapy. Hydrogels are hydrophilic three-dimensional (3D) highly porose materials, with very low apparent density in the dry state (<1 p/cm^3^) [49], capable of absorbing and maintaining a great amount of water by swelling and augmenting their dry volume hundreds of times [50]. The 3D network of hydrogels can form upon physical interactions, or upon covalent bonds formation by crosslinking reactions in solution or dry state [51]. Hydrogels can be produced using different synthetic and natural polymers, among which hyaluronic acid (HA) is a naturally occurring acidic mucopolysaccharide present in extracellular matrix (ECM), which has received extensive attention [52]. HA consists of a repeating unit of disaccharide, namely β-1,4-D-glucuronic acid—β-1,3-N-acetyl-D-glucosamine [53] (Figure 1).

HA per se owns curative effects to diverse diseases and has targeted features to cluster of differentiation 44 (CD44) receptors [54,55]. HA-based hydrogels are biodegradable and have thus been regarded as promising biomaterials to construct artificial organs, promote injury repair, and treat cancer [56]. Unfortunately, due to HA-based hydrogels’ sensitivity to hyaluronidase, if not properly reinforced by physical or chemical crosslinking, they are highly unstable, thus being inadequate for most biomedical applications. The physical crosslinking not involving new covalent bonds comprehends physical interactions, including electrostatic interactions, ionic exchanges, chain entanglement, hydrophobic self-assembly, hydrophobic/hydrophilic interactions, and hydrogen bonds [57,58]. Physical crosslinking could be preferable, because it normally leads to a rapid polymerization behavior under relatively mild conditions and needs no toxic crosslinkers or catalysts, thus decreasing potential cytotoxicity [59] and making them suitable to deliver drugs, encapsulate living cells and not only.

### 1.4. Present Study

Considering the above-mentioned premises, the aim of the present study is to find innovative antitumor molecules, possibly developable as new topical treatments for counteracting CMM; the anticancer effects of a BTPP-BA molecule have been investigated for the first time on two BRAF mutant’s melanoma cell lines, which commonly exhibit pigmentation on skin of the head, under the armpits, of the abdomen, and of the legs [60]. To this end, we selected the water-soluble sterically hindered quaternary phosphonium bola amphiphile salt (BPPB) shown in Figure 2.

As recent novelties, it was first reported to have potent antibacterial effects on a broad spectrum of MDR bacterial species and to possess excellent antiproliferative skills against sensitive and MDR neuroblastoma cells, regardless of their patterns of resistance. According to what was reported for TPP-based compounds, we have assumed it could act by a non-specific extra-genomic mechanism, probably involving membrane disruption [29,32]. Additionally, according to what is reported in the literature about the presence of the TPP group, in the case of neuroblastoma cells, a mechanism based on the detrimental accumulation of BPPB inside mitochondria, thus irreversibly impairing their functions and causing apoptosis, was hypothesized [29]. As observable in Figure 2, BPPB features two triphenyl phosphonium groups to target mitochondria, linked by a C12 alkyl chain. The possible cytotoxic activity of BPPB was here first assayed on patient-isolated MeOV and MeTRAV CMM cells, bearing BRAF^V600E^ and BRAF^V600D^ mutations, respectively. Then to have an early confirmation of BPPB capability to target mitochondria rather than specific mutations and impair their functions determining oxidative stress (OS), we evaluated its capability to induce ROS hyperproduction. Once the potential for BPPB as new active agent to be developed as a novel therapeutic to treat CM was established, supported by its selectivity for CMM cells and low cytotoxicity towards different non-tumoral mammalian cells, we formulated BPPB as hydrogel using hyaluronic acid (HA). The HA-based hydrogel loaded with BPPB could be used for the topical treatment of pigmentations caused by BRAF mutant CM and CMM cells, which usually appear on the skin of the head, abdomen, legs, and under the armpits [60]. HA should have served as a gelling agent by physical crosslinking reactions to achieve a dosage form for the intended dermal application. We thought that, owing to the anionic nature of HA in physiological conditions (Figure 1), specific positive ion and cationic molecules as BPPB capable of per se self-assembling in nanosized micellar structures could be suitable to fabricate a hydrogel, via electrostatic interactions, thus obtaining HA-based BPPB hydrogels without using possible toxic crosslinking agents.

## 2. Results and Discussion

### 2.1. 1,1-(1,12-Dodecanediyl)bis[1,1,1]-triphenylphosphonium di-Bromide (BPPB)

The TPP-based bola amphiphile compound, here biologically evaluated on MeOV and MeTRAV CMM cells, was synthetized according to Figure 1, performing the procedure previously described [32].

Although bola amphiphilic structures such as BPPB were never tested as anticancer agents, we were recently confident that the two TPP groups of BPPB and its colloidal properties could have allowed it to accumulate in mitochondria of cancer cells, disrupt the mitochondrial membrane potential, determine mitochondrial toxicity, and thus cause programmed death of cancer cells by inducing cell death processes (apoptosis, necroptosis, or autophagy) [17,33,36]. On these expectations, we recently successfully assessed the selective and potent cytotoxic effects of BPPB on both sensitive and MDR NB cells [29]. Since developing multitarget compounds active against several diseases with a low level of toxicity towards healthy cells could significantly reduce the costs of research, development, and production of novel therapeutic agents, we have reconsidered BPPB and assessed its cytotoxic effects on two CMM cells lines. To make the present study more relevant and original, we completed it by formulating BPPB as a hydrogel using HA for future possible topical treatment of CMM with reduced systemic side effects.

#### BPPB Characterization

Associated with a numbered structure of BPPB (Appendix A), the results from ATR-FTIR, ^1^H, ^13^C, and ^31^P NMR spectra, as well as those from FIA-MS-(ESI) and elemental analyses, have been included in Appendix A of our recent article [29]. Other useful information on BPPB characteristics have been included in Table S1 present in the Supplementary Materials associated with the same study [29].

### 2.2. Concentration- and Time-Dependent Cytotoxic Effects of BPPB on CMM Cells

We evaluated the cytotoxicity of BPPB on two human cutaneous metastatic melanoma (CMM) cell populations (MeOV and MeTRAV), derived from the biopsy of an untreated patient with cutaneous metastatic melanoma (CMM). Both cell lines are representative of the most lethal form of skin cancer, i.e., CMM, which show BRAF mutations in 50% of patients [61]. In this context, the identification of BRAF^V600E^ mutation in MeOV and BRAF^V600D^ in MeTRAV led to the development of specific orally available BRAF kinase inhibitors like PLX4032, developed by Plexxikon Inc. (Berkeley, CA, USA) and Hoffman-La Roche Ltd. (Basel, Switzerland) for the treatment of cancers harboring activating BRAF^V600E^ mutations [62]. Despite the initial success of these therapeutics, their clinical efficacy decreases after six months of therapy, leading to cancer relapse due to the onset of drug resistance. Therefore, developing new forms of treatment for CMM with limited or absent tendency to develop resistance is fundamental to improve therapy efficacy and reduce tumor recurrence. Both cell lines were exposed to BPPB in a concentration range 0.1–2.0 µM for 24, 48, and 72 h, selected based on the results obtained with BPPB on neuroblastoma cells [29].

#### 2.2.1. Cytotoxic Effects of BPPB on CMM Cells

As reported in Figure 3 (MeOV cells) and Figure 4 (MeTRAV cells), BPPB caused a significant reduction in cell viability starting from the lowest concentration used (0.1 µM). Both concentration- and time-dependent cytotoxic effects were observable, especially evident on MeOV cells, which demonstrated slightly less tolerance to BPPB with respect to MeTRAV CMM cells. While in both populations a time-dependent effect was observable passing from 24 to 48 h of exposure, the BPPB cytotoxicity remarkably increased when cells were exposed for 72 h. Concerning the more sensitive MeOV CMM cells, while to decrease cell viability under 50% (49.3%), BPPB 1 µM was necessary after 24 h treatment, and a concentration 0.25 µM was needed for the 48 h treatment (46.1%), after 72 h of exposure, the lowest concentration of 0.1 µM was sufficient to practically kill all cells (27.7% of live cells).

Although MeOV are metastatic CM cells, MeTRAV CMM cells demonstrated more tolerance to BPPB exposure than MeOV ones. While cell viability decreased under 50% at BPPB 1.5 µM after the 24 h treatment (45.0%), and a concentration 0.75 µM was needed in the 48-h treatment (41.5%), after 72 h of exposure, the lowest concentration of 0.1 µM was sufficient to reduce cell viability at 46.6%. However, to reduce viability of the MeTRAV cell population to 26.0%, as observed in the case of MeOV at concentration 0.1 µM of BPPB (27.7%), an amount of BPPB 7.5 times higher was necessary after 72 h treatment. Curiously, in the treatment at 48 h, for BPPB concentrations ≥ 0.75 µM cell viability of MeTRAV was slightly lower than that of MeOV. Collectively, at the very low highest concentration of BPPB used in these experiments (2.0 µM) viability of cells was insignificant both after 48 and 72 h treatment (14 and 5% for MeOV, 10 and 7% MeTRAV).

#### 2.2.2. Determinations of the IC_50_ Values and of Selectivity of BPPB for CMM Cells vs. Non-Tumoral Cells

##### IC_50_ Values

From results reported in the previous Section 2.2.1, very low values of IC_50_ could be expected, especially after 72 h of treatment. To determine the IC_50_ values of BPPB on both cell populations and at all times of exposure here considered, we first converted the bar graphs in Figure 3 and Figure 4 into dispersion graphs (Appendix A, Appendix A associated to this article). Then, upon transformation of BPPB concentrations in Log_10_ concentrations (Appendix A, lines with error bars and indicators) and by fitting a nonlinear model to the obtained data, the plot of Log_10_ concentrations vs. the normalized response was achieved (Appendix A, lines without either error bars or indicators), which was used to calculate the IC_50_ values reported in Table 1.

Data reported in Table 1 evidence that, as expected, a very strong time-dependent effect on cytotoxicity of BPPB actually exists against both types of cells. Particularly, although very low IC_50_ values were observed already after the first 24 h of exposure towards both MeOV and MeTRAV CMM cells, and especially towards MeOV CMM cells (0.84 µM vs. 1.18 µM), on MeOV cells, the IC_50_ values at 48 h were 2.5-fold lower than at 24 h of treatment. Also, after 72 h of exposure, the IC_50_ values were 6.8-fold lower than those detected at 48 h and 16.9-fold lower than those noticed at 24 h. A similar, even if soft scenario, due to the higher tolerance of these cells to BPPB, was evident also for MeTRAV CMM cells. After 48 h of exposure, they demonstrated IC_50_ values 2.8-fold lower than those noticed at 24 h. Further, after 72 h of exposure, the IC_50_ values decreased of additional 2.4-fold with respect to the values detected after the 48-h treatment and were 6.9-fold lower than those observed after the first 24 h of exposure. The following Figure 5 visualizes the decreasing of the IC_50_ values of BPPB on both cell populations as a function of the time of treatment.

Considering our previous work on CMM, where MeOV and MeTRAV CMM cells were subjected to intensive administration of PLX 4032 to select their resistant counterpart for better studying the molecular mechanisms of resistance emergence to PLX4032 [61], the reported IC_50_ values of PLX4032 after 72 h treatment were 0.5 µM and 18.7 µM towards MeOV and MeTRAV, respectively. The high difference in sensitivity of these two cells populations to this drug is due to the specificity of PLX4032 for counteracting CMM cells with BRAF^V600E^ mutation as MeOV, rather than BRAF^V600D^ as MeTRAV [61]. In this regard, BPPB has demonstrated cytotoxic effects more potent than those of PLX4032 by 10 times on MeOV CMM cells and by 110 times on MeTRAV CMM cells, thus establishing that the possible use of BPPB to treat CMM could remarkably reduce the difference of sensitiveness existing between cell lines, probably acting by an extra-genomic mechanism, not involving the specific BRAF mutations of the different CMM cells. While when administered with PLX4032, MeTRAV cells were less tolerant than MeOV cells to the drug by 37 times, in case of BPPB administration, they were more tolerant than MeOV cells by only 24 times. Additionally, probably supported also by its nanosized self-formulation, much lower concentrations of BPPB were necessary to inhibit both cell populations, with respect to PLX4032, known to select CMM cells resistant also to drugs different from PLX4032, after about six months of treatment, causing tumor recurrence [61]. This feature is of paramount importance because it could significantly reduce the possibility of selecting a multidrug resistant form of both cell lines.

##### Selectivity Indices

A sufficiently high value of the selectivity index (Equation (1)) is an essential requirement to render a new molecule worthy of consideration for further studies and future development as new therapeutic agent.
(1)$$SI=\frac{{IC}_{50}NTC}{{IC}_{50}CMM}\;$$

To obtain an acceptable *SI*, the *IC_50_* values on non-tumoral cells (*NTC*) should be as high as possible compared to the *IC*_50_ values determined for tumor cells in or *CMM* cells.

The cytotoxic effects of BPPB on several different types of non-tumoral mammalian cells and its hemolytic toxicity on red blood cells (RBCs) have been recently reported and are summarized expressed as IC_50_ in Table 2 [29]. Mammalian cells included non-immortalized human MRC-5 lung fibroblasts, as well as immortalized human hepatic HepG2 cells and monkey kidney Cos-7 cells, and RBCs from blood of four volunteer healthy donors [29]. The IC_50_ values determined here for both MeOV and MeTRAV MM cells were also included in Table 2 for a direct comparison.

The IC_50_ values reported in Table 2 determined after 24 h of exposure and those determined on RBCs after the time of experiments based on existing protocol [29] are also visualized in Appendix A. According to the results in Table 2, the IC_50_ values of BPPB on both CMM cell populations were significantly to exceptionally lower than those determined on the non-tumoral mammalian cells considered, which practically embraced the most representative cells and organs (lung, kidney, liver, and blood) with which a possible therapeutic could interact after topical administration and upon adsorption. The SIs were then calculated according to Equation (1) and are reported in Table 3. Considering the exposure timing of 24 h, concerning that data are available on all eukaryotic cells tested, a SI range of 2.35–299.00 was evidenced.

To further evidence how the anticancer effects of BPPB strongly depend on the exposure timing and how this phenomenon is significantly less evident when BPPB is used on non-tumoral cells, the values of selectivity indexes obtained considering MRC-5 cells (as a model of non-tumoral and non-immortalized cells) and RBCs were reported in graphs as a function of time of exposure (Figure 6).

As observable in Figure 6A, although cytotoxicity of BPPB vs. MRC-5 cells significantly increased with increasing times of exposure, the remarkably major extent to which the cytotoxic effects of BPPB towards CMM cells increased over time, in 72 h treatment experiments, allowed SI values 3.1-fold (MeTRAV) and 8.5-fold (MeOV) higher than those observed in 48 h treatment experiments. The phenomenon was even more evident when RBCs were considered (Figure 6B). SIs calculated from BPPB IC_50_ values on MeOV and MeTRAV CMM cells treated for 48 h were 2.5- and 2.8-fold higher than those calculated from BPPB IC_50_ values on MeOV and MeTRAV CMM cells treated for 24 h, respectively. When SIs calculated from BPPB IC_50_ values on MeOV and MeTRAV CMM cells treated for 72 h were considered, they were 6.8- and 2.4-fold higher than those calculated from BPPB IC_50_ values on MeOV and MeTRAV CMM cells treated for 48 h, respectively, and 16.9- and 6.9-fold higher than those calculated from BPPB IC_50_ values on MeOV and MeTRAV CMM cells treated for 24 h, respectively.

### 2.3. Concentration- and Time-Dependent ROS Induction by BPPB on CMM Cells

Although previous studies reported that TPP-based compounds, including TPP-BA as BPPB, exert their biological effects mainly targeting and impairing mitochondria functions with consequent high hyperproduction of ROS and oxidative stress (OS) [29,32,36], a preliminary investigation of the possible ROS-dependent mechanism at the base of the potent anticancer effects of BPPB on MeOV and MeTRAV CMM cells was assessed. To this end, we simply evaluated the possible improvement of production of ROS in both cell populations when treated with the same BPPB concentrations and for the same timing as performed in the cytotoxicity experiments. By reporting the BPPB concentrations used vs. the ratio DCFH positive cells (%)/viable cells (%), the bar graphs in Figure 7 and Figure 8 were obtained for MeOV and MeTRAV CMM cells, respectively.

As shown in Figure 7 and Figure 8, BPPB strongly increased H_2_O_2_ levels in a time- and dose-dependent manner, in both cells’ populations, although the effect of timing on ROS production was particularly evident, especially at 72 h of treatment. This trend was like that observed in the cytotoxicity experiments, also envisaging an actual ROS-induced cytotoxicity and a strong correlation between ROS induction and cell death. Thus, to confirm these two empirical assumptions, we first reported the ratios DCFH positive cells (%)/viable cells (%) observed upon administration of the max concentration of BPPB (2.0 µM) to both cells’ lines, vs. exposure timing (Figure 9), in a dispersion graph to visualize if a dependance of ROS improvement by time of treatment actually existed.

Unequivocally, ROS hyperproduction, induced by BPPB on both cell populations at the maximum concentration tested (2.0 µM) chosen as reference, was time-dependent, as asserted for BPPB cytotoxic effects. Specifically, the values of the ratios DCFH positive cells (%)/viable cells (%) increased by 3 times and 4 times, respectively, passing from 24 to 48 h, then to 72 h exposure, when MeTRAV CMM cells were considered, and by 2 times and 5.8 times, respectively, when MeOV CMM cells were tested. Secondly, to confirm our second hypothesis of a correlation existing also between cytotoxicity effects of BPPB and its capability to induce ROS hyperproduction, we reported the ratios DCFH positive cells (%)/viable cells (%) vs. cell viability (%) in logarithmic scales on both x and y axes. The logarithmic scales, selected as mathematical elaborations of the original data, allowed to visualize the degree of linearity of the possible linear regressions built on the obtained dispersion diagrams by evaluating their coefficient of determination (R^2^) values. High values of R^2^ would have confirmed the assumed correlation. Figure 10 shows the results of such an operation considering both MeOV (Figure 10A) and MeTRAV (Figure 10B) cells.

As expected, the high R^2^ values in the range 0.95–0.99 for MeOV CMM cells (Figure 10A) and those in the range 0.96–0.99 for cells MeTRAV (Figure 10B) confirmed that an inverse exponential correlation between ROS hyperproduction and cell viability (%) exists in both cases. Figure 10 evidenced that the increasing of ROS corresponded to an increase of cell death, thus envisaging a possible cell death by oxidative stress (OS) causing apoptosis by mitochondrial impairments caused by interactions with BPPB. The correlation seemed to be lightly stronger for MeTRAV cells, and in both cases seemed to be lightly stronger for 24 and 48 h of exposure than for 72 h.

### 2.4. Hyaluronic Acid (HA)-Based BPPB Hydrogel (HA-BPPB-HG)

Despite several protocols having been reported in recent years for preparing empty or drug-loaded HA-based gels and nanogels using different polymers, and by both chemical and physical methods to induce jellification, they appeared too complex to us [63,64,65]. Most needed a preliminary chemical modification of HA before gel formation, mainly using cholesterol [66] or other molecules, including boronic acid esters [67]. Aiming at preparing HA-based hydrogels loaded with BPPB in a method as simple and low-cost as possible, we attempted to formulate BPPB as a hydrogel using unmodified HA and carrying out a procedure like those recently reported [68,69]. A similar method was also successful in formulating an imidazo-pyrazole (4I) as hydrogel (R4HG-4I) with excellent physicochemical properties, using a synthesized polystyrene-based resin (R4) [50]. In this specific study, the 3D network of R4HG-4I, simply self-formed upon dispersion of the two ingredients in water by physical interactions including acid–base interactions and hydrogen bond formations between the cationic resin and the basic nitrogen atoms of 4I. Here, the 3D hydrogel network could have formed, through similar physical connections, precisely ionic interactions, between the negative charges of hyaluronate and the positive ones of cationic nanoparticles (NPs), that BPPB has been shown to form in aqueous solution. By this technique, the gel (HA-BPPB-HG) was obtained by simple dispersion of the only two ingredients in excess water, without needing of particular high temperatures or instruments, such as autoclaves, preliminary functionalization, or of using potentially toxic crosslinking agents [70]. The as-prepared HA-BPPB-HG hydrogels from five independent reactions (average weight ± SD = 160.6 ± 1.1 mg), were achieved containing the maximum amount of water they could absorb and at their maximum degree of swelling. The water absorption capacity (WAC (%)) of the just-prepared hydrogel (HA-BPPB-HG) when water was absorbed during the 3D network formation was calculated by Equation (1). The WAC (%) was 1238.3%, evidencing its high capability of retaining water. This was in accordance with WAC (%) determined by experiments reported in Section 2.4.5 (1261.8%), with a +1.9% difference in the second case, probably due to the higher time of contact with water allowed to the fully dried and already formed and stabilized HA-BPPB-HG 3D network used in the following experiments. The not-entrapped BPPB remained dissolved in the not-absorbed water, which was separated by centrifugation and dried providing an oily residue. Upon ATR-FTIR analyses, its identity with BPPB was confirmed (Appendix A). The soaked HA-BPPB-HG from five different reactions were lyophilized, obtaining 11.3, 10.8, 10.6, 11.2, and 11.0 mg (10.98 ± 0.26) of fully dried gels which were stored separately in a drier, due to their high hygroscopicity. Once characterization was completed, the hydrogels from the five preparations were collected using water, then freeze-dried to achieve about 50 mg of foamy/glassy withe solid (Appendix A).

#### 2.4.1. ATR-FTIR of HA, BPPB, and HA-BPPB-HG

ATR-FTIR analyses were carried out on HA, BPPB, and fully dried HA-BPPB-HG, directly on the solid samples (Figure 11, Appendix A).

At first observation, the spectrum of HA-BPPB-HG appeared identical to that of HA and bands belonging to BPPB were unlikely to be detected despite the high loading (DL%) spectrophotometrically determined (3%) in the subsequent Section 2.4.2. and the initial cargo. Briefly, bands belonging to the C=O stretching were observable at 1602 and 1604 cm^−1^ in the spectra of HA and HA-BPPB-HG, respectively, as well as those of C-O stretching at 1027 and 1028 cm^−1^. In place of the band of methyl/methylene groups of saccharide unites of HA (2889 cm^−1^), it was possible to appreciate the band at 2924 cm^−1^ peculiar to BPPB in the spectrum of HA-BPPB-HG. Moreover, bands at 1321 and 689 cm^−1^ were observable, probably due to the aromatic and aliphatic C-P stretching of BPPB.

##### PCA Results and Discussion

To have a more reliable information on composition of HA-BPPB-HG, we processed spectral data by Principal Components Analysis (PCA), using CAT statistical software as reported in the Experimental Section. These operations allowed us to reduce the large number of correlated variables (10,203), bearing also a lot of inoperable information, into three non-correlated variables (namely Principal Components PC1, -2, and -3) bearing only the most useful information. Results have been reported in the form of score plot of a selected component (PC1, -2, or -3) vs. another of the residual ones, representing a new space in which samples are located with respect to others according to their chemical composition characteristics, and scores are their new coordinates in such new space. The following Figure 12 represents the score plot of PC1 vs. PC2, while Appendix A is that of PC2 vs. PC1. In this case, PC3 was excluded and not further considered because 100% of the total variance was already explained by PC1 and PC2 (Figure 12 and Appendix A).

For PC1 (90% of variance), the location of samples, with HA and HA-BPPB-HG both placed at a positive score > 20, evidenced a strong component given by HA in HA-BPPB-HG, as already disclosed in the FTIR spectra, although the initial cargo. This phenomenon could be explained by the peculiar major intensity of some saccharides’ bands in HA (C=O, C-O) despite their concentration, with respect to those typical of BPPB and characteristically less intense. However, the location of the samples on PC2 (10% of total variance) showed BPPB and HA-BPPB-HG placed both at a positive score in the range 0–20 and HA located distant from them at a negative score (−20), unequivocally evidencing the presence of BPPB in HA-BPPB-HG, according to the initial cargo and DL% assessed in the following Section.

#### 2.4.2. Assessment of BPPB Content in HA-BPPB-HG

Upon construction of the BPPB calibration curve whose linearity was confirmed by the very high (>0.99) coefficient of determination R^2^ (Appendix A), the amount of BPPB contained in an exactly weighted sample of HA-BPPB-HG was determined by spectrophotometric UV-Vis analyses by reading the absorbance (Abs) at 267 nm. Table 4 summarizes the results.

According to results reported in Table 4, HA-BPPB-HG was obtained with exceptionally high drug loading (DL%) and high encapsulation efficiency (EE%) values, which have been included in Table 4. Differently from results obtained previously when adopting this protocol [50], while the EE% was only slightly lower to that previously observed, the DL% measured this time was 9.8 times higher. We can find the reason for this remarkable difference in the significantly stronger electrostatic interactions which were possible between the anions of HA and the cations of BPPB, rather than the weaker acid–base interactions which were possible between imidazo-pyrazole (4I) and protonated resin (R4HG). The ionic interactions which were possible this time promoted a higher loading of the BPPB vesicles, also supported by the highly porous structure of the formed 3D network, as it was revealed by SEM investigations (see the following Section 2.4.3) which endorsed the accommodation of BPPB nano vesicles inside the hydrogel.

#### 2.4.3. Scanning Electron Microscopy (SEM)

The SEM images of HA-BPPB-HG revealed an interconnected porous microstructure (Figure 13A,B), presenting a so-called honeycomb porous construction, especially visible in Figure 13B. The honeycomb-type structure reproduces that of the natural extracellular matrix (ECM). The obtained pore size distributions demonstrated that the structure of HA-BPPB-HG was relatively inhomogeneous, with larger pores existing close to smaller ones. The average pore sizes were determined to be about 271 µm, with pore sizes ranging between approximately 25 µm and 750 µm. As specified in Section 2.4.5, this high porous structure translated to a very high porosity percentage (98%) The here-fabricated microporous structure of HA-BPPB-HG, as well as its high porosity (%), helped the entrapment of BPPB, which translated into the high values of DL (%) and EE (%) reported in Section 2.4.2 (Table 4).

#### 2.4.4. Equilibrium Swelling Rate

Swelling determinations were made at fixed time points as described in the Experimental Section and reproducing the scheduled times previously reported [71,72], at different pH values of 4, 7, and 10. All experiments were performed in triplicate, and results were expressed as mean ± SD. The swelling experiments provided the samples in the form of soaked hydrogels. The cumulative swelling rate (%) over time was calculated and the obtained data were plotted vs. time, obtaining the curves in Figure 14.

The tested samples of HA-BPPB-HG swelled dynamically and practically to the equilibrium swelling degree (ESD%) during the initial first hour (24 min). This early rapid swelling occurring already during the first 15 min and translating into a swelling (%) of 950% (pH = 7), 1100% (pH = 4), and 690% (pH = 10) was due to the porous and sponge-like structure of HA-BPPB-HG, which facilitated the diffusing of water molecules into the hydrogels, as Gulfam et al. reported previously [73]. A slight dependence of such diffusion on pH was observable, with the process promoted by an acidic pH and limited by a basic one. The equilibrium swelling rate (*Qe*) was determined at the time point over which the swelling rate (%) did not further improve and a constant weight was detected. Graphically, *Qe* values corresponded to the intercept on y axis of the tangents to the plateau of curves in Figure 14 and were the same in all conditions (1250%), as indicated in Figure 14. This finding underlines how the swelling properties of HA-BPPB-HG were more dependent on gel porosity rather than on the pH of the swelling media. On the contrary, the time to reach the experimental *Qe* value was different in the three different conditions. Particularly, *Qe* was reached after 60, 75, and 120 min in acidic, neutral, and basic solutions, respectively. Compared to other hydrogels reported to be promising for biomedical application, samples of HA-BPPB-HG demonstrated significantly higher *Qe* values [74]. Particularly, HA-BPPB-HG showed a maximum swelling about three times higher than that showed by the best sample reported in the study.

##### Kinetic Studies

The kinetics governing the water adsorption by HA-BPPB-HG were assessed by carrying out a kinetic study, as previously reported [75,76,77]. To this end, the data of the cumulative swelling ratios plotted in Figure 14 were fit with kinetic models of pseudo-first order (PFO) (Equation (2)) and pseudo-second order (PSO) (Equation (3)).
(2)$$Ln\left(Qe\;\left(\%\right)-Qt\left(\%\right)\right)=lnQe\;\left(\%\right)-K\left(PFO\right)\times t$$
(3)$$\frac{t}{Qt\;(\%)}=\frac{1}{K(PSO)\times Q{e\;(\%)}^{2}}+\frac{1}{Qe\;(\%)}t\;$$
where *Qe* (%), and *Qt* (%) are the swelling (%) at equilibrium and at time *t*, respectively, *K*_(PSO)_ is the constant of the PSO kinetic model, and *K*_(PFO)_ is the constant of the PFO kinetic model. Values of ln(*Qe* − *Qt*), and *t*/*Qt* were plotted vs. times. The obtained dispersion graphs were processed by Microsoft Excel software 365 using the Ordinary Least Squares (OLS) method, and their linear regression lines were obtained. As reported, the coefficients of determination (R^2^) of their equations were the parameters for determining the kinetic model that best fit the experimental data of the water absorption processes [77]. In all conditions, HA-BPPB-HG samples best fit the PSO kinetic model. *K*_PSO_ and *Qe* (%) from the model (*Qe* (%)_PSO_) were obtained by the intercept and slope, respectively, of equations reported in Appendix A, and are included in Table 5.

*Qe* (%)_PSO_ demonstrated the exact coincidence between the experimental *Qe* (%) (*Qe* (%)_EXP_) and those calculated by PSO model (*Qe* (%)_PSO_), thus confirming that the PSO equation is suitable to model the obtained absorption experimental data. The values of K evidenced that swelling was more rapid at acidic pH, followed by adsorption in the basic and neutral solutions.

##### Rheologic Considerations

It is reported that a reorganization of polymer (HA) chains which pass from a more compact glassy state to a less compact gummy state occurs during swelling. The differences in the appearance of HA-BPPB-NG can be appreciated by comparing the images in Appendix A, which represent HA-BPPB-HG in its glassy state, with those in Appendix A, which represent HA-BPPB-HG in its gummy state. The time required for this process depend on the relaxation time (λ_m_) which is typical for the considered system polymer/solvent, in our case, HA-BPPB-HG/water. If such reorganization is enormously more rapid (very high value of λ_m_) or extremely slower (very low value of λ_m_) than the time of diffusive phenomena (θD), the adsorption, and therefore the swelling, will be ruled by Fickian diffusion [78]. On the contrary, if λ_m_ has a value close to that of the diffusive phenomenon, the swelling is no longer more ruled by Fickian diffusion but by the relaxation itself, and is defined anomalous [79]. When Fickian diffusion governs the swelling of a system, it could behave as a viscous system or as an elastic one. Particularly, when λ_m_ is very low, the system behaves as a liquid and is defined viscous, while when λ_m_ is very high, the system behavior is more assimilable to that of a solid and it is defined elastic. When λ_m_ ≃ θD [78], the swelling is odd, and the system is defined typically viscoelastic [79]. Although we did not know the values of both λ_m_ and θD, the kinetic model (PSO) that fits the swelling of HA-BPPB-HG in all conditions has evidenced that Fickian diffusion is not applicable to our system, which is considerable a viscoelastic one. To give context, two videos (Appendix A) are available showing the slow descending of HA-BPPB-HG on vertical glass walls and its slow return to the original condition once deformed (Video S1) and its slow deformation from the original condition visible in central image in Appendix A.

#### 2.4.5. Water Absorption Capacity (WAC (%)), Equilibrium Water Content (EWC (%)), and Porosity (%)

The WAC (%) of HA-BPPB-HG, defined as its possible maximum swelling, its equilibrium water content (EWC (%)), and porosity (%), defined as the void spaces fillable with water during swelling, were determined as described in Section 2. All experiments were performed in triplicate, and results were expressed as mean ± SD (Table 6).

WAC (%) measured here were slightly higher (1262% vs. 1250%, +0.95%) than the maximum swelling (%) obtained as equilibrium swelling rate (*Qe*) and by +1.9% higher than WAC (%) determined on the just-prepared hydrogel. We have already justified the second case by supposing that, while still in formation, the 3D network of HA-BPPB-HG could have a reduced capability of absorbing water; once complete, due to its high porosity (98%), its aptitude to soak water could be significantly augmented. On the other hand, we assumed that EDS (%), also defined by *Qe* (%), could be less than WAC (%), because while WAC (%) refers to an experiment where HA-BPPB-HG was in contact with excess of water for a continuative time of 24 h, EDS (%) was measured over a time during which the contact of HA-BPPB-HG with water was interrupted at fixed time points during which water was removed to be replaced with new aliquots. Concerning WAC (%), HA-BPPB-HG demonstrated a capacity to absorb water higher than that of two out of four nanocellulose fibers (NCFs)/collagen aerogels which proved capacity in supporting cell growth and proliferation [80], and identical to that of an aerogel-like compound (M4) recently proposed as scaffold for regenerative medicine [81]. HA-BPPB-HG cannot be defined an aerogel as M4 because both its porosity and its apparent density value (0.52 g/cm^3^) did not respect the required ones. An extremely low density would require, which, by definition, should be in the range 0.35–0.5 g/cm^3^ [82,83]. Concerning EWC (%), values higher than those of FRH-PG and FRH-PGS scaffolds have been detected [84]. Porosity is a crucial parameter for hydrogels used in tissue engineering, wound healing, drug delivery, and other biomedical applications. It reflects the interconnected void spaces within the hydrogel structure, affecting properties like water/exudate absorption, drug loading, and mechanical behavior.

Collectively, it has been reported that 3D networks with 60–90% porosity are suitable for biomedical applications, as they can provide sufficient space for cell activity, oxygen and nutrient exchange, and the production of a new extracellular matrix (ECM) [85]. In this regard, HA-BPPB-HG demonstrated a porosity close to the highest value of this range, thus confirming its suitability for dermal application also onto lesioned skin. Additionally, the antibacterial activity of BPPB contained in the hydrogel recently reported could help in case of infected CMM lesions [32].

#### 2.4.6. In Vitro Evaluation of Biodegradability of HA-BPPB-HG over Time by Mass Loss Experiments

In these experiments we investigated the degradation of HA-BPPB-HG at pH = 4, 7, and 10. Regardless the pH value, the degradation of HA-BPPB-HG was due to the progressive absorption of water, which first induced swelling within the first 60 min as observed in the swelling experiments and secondly began to interfere with the ionic interactions existing between HA and BPPB. This phase was followed by the progressive hydrolysis of the linear chains of hyaluronic acid (HA), with consequent copious release of BPPB (see release experiments). Figure 15 shows the three mass loss profiles.

As observable in Figure 15, the max percentage of degradation was over 80% and very similar in all cases with the highest one (86%) reached at acidic pH after 72 h (3 days), followed by 84% and 81%, reached at basic and physiological pH after a longer time of 144 h (6 days). Collectively, HA-BPPB-HG samples at acidic pH degraded to a major extent, and degradation was significantly more rapid than at basic and physiological pH, possibly establishing a more rapid and quantitative release of BPPB. At basic pH, the release was practically linear until 96 h, then decreased before reaching a plateau during the next time period. At physiological pH, the biodegradation profile was more complex, alternating linear tracts of rapid degradation to plateau tracts (Figure 15). The pH of cutaneous melanoma lesions can vary, but they tend to be more acidic compared to normal skin. This acidity is often due to the metabolic activity of the cancer cells, which produce lactic acid as a byproduct of anaerobic glycolysis [86]. Additionally, in a recent work, the glycolytic metabolism via production of lactic acid was demonstrated for the MeOV and MeTRAV CMM cells used here [61]. This lower pH environment can influence tumor progression and the effectiveness of certain treatments [86]. In our case, the acidic environment of CMM lesions could promote a rapid degradation of HA-BPPB-HG and the quantitative release of BPPB within three days, the time necessary to it to induce strong ROS production and to fully exterminate CMM cells at the lowest concentrations of 0.05 and 0.17 µM. The biodegradability of our samples was compared with that of the multifunctional carboxymethyl chitosan and oxidized cellulose biodegradable (COB) hydrogel scaffolds developed by Shengyu Li et al. [87], who monitored the weight loss of their samples for only four days. The maximum biodegradation of HA-BPPB-HG was higher than that of all COB hydrogels at all explored conditions [87].

#### 2.4.7. Evaluation of BPPB in Vitro Release over Time

The in vitro release over time of BPPB from HA-BPPN-HG was assessed spectroscopically (UV-Vis) at 267 nm using the dialysis membrane method according to the literature [60]. Particularly, the experiments were performed in a physiological medium (PBS, pH 7.4) and at pH = 4 at temperature incubation of 37 °C, to simulate the two possible conditions of a cutaneous melanoma at its first stage (pH of tumor environment close to neutral) and of a CMM in its advanced stage (pH of CMM lesion more acidic). The concentrations of BPPB released in the media were used to compute the cumulative drug release percentage over time. The obtained data were plotted vs. time, obtaining the curves indicating the BPPB release profiles shown in Figure 16.

As expected, after having observed the curves of degradation, the release of BPPB from the hydrogel was more rapid in an acidic pH than in a neutral one, confirming the assumption that hydrolysis of HA helped its release. At the very early beginning, the BPPB release was rather independent from HA degradation, and instead more dependent on its rapid swelling, enlargement of the 3D network and the initial breaking of the ionic interactions between the anionic HA and cationic BPPB. In fact, a strong burst release was observed within the first four hours in both cases, with a 77% release at pH = 4 and a 35% release at pH = 7 when degradation was still minimal, about 15% and <5%, respectively. During the next timing period, the release of BPPB at pH = 4 went on to reach completion (99.2%) after 3 days, when degradation reached its maximum of 86%. On the contrary, at neutral pH, the release of BPPB reached the 75% after 28 h when degradation was slightly over 20%, proceeding slowly up to 90% after 3 days and remaining steady. Collectively, even if an initial strong release of BPPB exists, the release profiles in both conditions evidence the capability of HA-BPPB-HG of allowing sustained and protracted release of BPPB vs. time. In this regard, the release profile of BPPB from HA-BPPB-HG was significantly more controlled and sustained than that observed by Wang et al. for their fluorouracil (FU)-loaded pH-sensitive konjac glucomannan/sodium alginate (KGM/SA) and KGM/SA/graphene oxide (KGM/SA/GO) hydrogels [88].

##### Kinetic Studies

To better understand the kinetics and the main mechanisms which govern the release of BPPB, as previously described for other experiments, the data plotted in Figure 16 were fitted to the most common kinetic models [50,69,89]. The obtained dispersion graphs were fit with the related linear regressions built up by Microsoft Excel 365 software using the OLS method, and the highest value of their coefficients of determination (R^2^) was considered as the parameter to determine which model better fits the release data. Accordingly, the release of BPPB from HA-BPPB-HG best fit with the PSO model both when assessed at pH = 4 and at pH = 7 (Appendix A), whose Equation for this case is the following (Equation (4)).
(4)$$\frac{t}{D(\%)t}=\frac{1}{K(PSO)\times D\left(\%\right)\;e^{2}}+\frac{1}{D\left(\%\right)\;e}t\;$$
where *D* (%) *t* is the amount of BPPB released at time *t*, *K*_(PSO)_ is the PSO kinetic constant, *t* is time, and *D* (%) *e* is the amount of drug (BPPB) released at the equilibrium, i.e., the maximum drug release (%). Using Equation (3) and the equation of the linear regression reported in Appendix A, we calculated both the *K*_(PSO)_ and the maximum drug release *D* (%) *e* for both the experimental conditions using the values of intercept and slope respectively. The values are reported in Table 7, and the calculated *D* (%) *e* was compared to the experimental one to establish the good fit of PSO model with the BPPB release data.

For Table 7, the maximum drug release computed using the model sufficiently agreed with the experimental values, confirming that the PSO kinetic model well describes the mechanism governing the BPPB release from HA-BPPB-HG. In systems of type-loaded polymers/aqueous media ruled by PSO kinetics, chemical processes involving the sharing or exchange of electrons between the system and water or electrostatic interactions are the main mechanism by which water is adsorbed and a solute can release molecules possibly loaded [68,76], which confirms our previous empirical assumptions.

Differently, Fickian diffusion is the main mechanism occurring for systems ruled by Higuchi kinetic model, similar to the releases of FU from pH-sensitive konjac glucomannan/sodium alginate (KGM/SA) and KGM/SA/graphene oxide (KGM/SA/GO) hydrogels reported by Wang [88] or those previously reported by us [50].

## 3. Materials and Methods

### 3.1. Chemicals and Instruments

Hyaluronic acid (HA, MW = 6.4 KDa) was purchased from Lifescore, BIOMEDICAL, Chaska, MN, USA. All other reagents and solvents used in this study were obtained from Merk (Milan, Italy) and were used without further purification. 1,1-(1,12-dodecanediyl)*bis*[1,1,1]-triphenylphosphonium di-bromide (BPPB) was synthetized and characterized as recently described [32]. All analyses performed to characterize BPPB were carried out on the instruments and with procedures previously reported [32]. A peaks/bands list by ATR-FTIR, ^1^H, and ^13^C NMR, including DEPT-135 experiments, as well as ^31^P NMR spectroscopic analyses, is available in the Appendix A associated with a recent article [29], while their images, as well as those of other interesting analytical characteristics, are included in Appendix A of the same publication [29].

### 3.2. BPPB Cytotoxicity Evaluation on CMM Cells

#### 3.2.1. Cell Culture Conditions

MeOV and MeTRAV cell lines, directly obtained from the biopsy of an untreated patient with cutaneous metastatic melanoma (CMM), were kindly provided by Prof. Gabriella Pietra (University of Genoa, Genoa, Italy). The authenticity of the selected cells was checked by Short Tandem Repeat (STR) profile analysis performed by the Immunohematology and Transfusion operative unit, IRCCS Ospedale Policlinico San Martino, Genoa. Both cell lines were maintained in RPMI 1640 medium (Euroclone Spa, Pavia, Italy) supplemented with 10% Fetal Bovine Serum (FBS, Euroclone Spa, Pavia, Italy), 1% L-Glutamine (Euroclone Spa, Pavia, Italy), and 1% Penicillin/Streptomicin (Euroclone Spa, Pavia, Italy), and grown in standard conditions (37 °C humidified incubator with 5% CO_2_).

#### 3.2.2. Treatments

To determine the cytotoxic effects of BPPB, time- and dose-dependent experiments were carried out. Cells were first treated for 24, 48, and 72 h with increasing concentration of BPPB in the range 0.1–2.0 µM. The stock solutions of these compounds were prepared in 40,000-fold diluted DMSO, and pilot experiments demonstrated that the final DMSO concentrations did not change any of the cell responses analyzed. Cell cultures were carefully monitored before and during the experiments to ensure optimal cell density. Notably, samples were discarded if the cell confluence reached >90%.

#### 3.2.3. Cell Viability Assay

Cell viability was determined by using the CellTiter 96^®^ AQueous One Solution Cell Proliferation Assay (Promega, Madison, WI, USA), as previously described [90,91]. Briefly, cells (10,000 cells/well) were seeded into 96-well plates (Corning Incorporated, Corning, NY, USA), then treated. Next, cells were incubated with 20 µL of CellTiter, and the absorbance at 490 nm was recorded using a microplate reader (EL-808, BIO-TEK Instruments Inc., Winooski, VT, USA). The cell survival rate, expressed as cell viability percentage (%), was evaluated based on the experimental outputs of treated groups vs. the untreated groups (CTR) and was calculated as follows: cell viability (%) = (OD treated cells − OD blank)/(OD untreated cells − OD blank) × 100%. IC_50_ was evaluated by GraphPad Prism 5.4.2 Software (GraphPad Software, Boston, MA, USA) as explained in the Discussion Section.

#### 3.2.4. Detection of Hydrogen Peroxide (H_2_O_2_) Production

The production of H_2_O_2_ was evaluated using 2′-7′-dichlorofluorescein-diacetate (DCFH-DA; Merk Life Science S.r.l., Milan, Italy), as previously reported [92].

#### 3.2.5. Statistical Analyses

Results have been expressed as means ± S.D. of four independent experiments in which different wells were analyzed every time for each experimental condition. In the analyses of cell viability or H_2_O_2_ levels, the condition of untreated cells was set as 100% ± SD and 2 ± SD, respectively. Statistical significance of differences was determined by one-way analysis of variances (ANOVA) followed by Dunnet’s multiple comparison test correction using GraphPad Prism 8.0.1 (GraphPad Software v8.0, San Diego, CA, USA). Asterisks indicate the following *p*-value ranges: * *p* < 0.05, ** *p* < 0.01, *** *p* < 0.001, **** *p* < 0.0001. *p* > 0.05 was not considered statistically significant and no symbol was used in the images.

### 3.3. Preparation of Hyaluronic Acid (HA)-Based BPPB Hydrogel (HA-BPPB-HG)

A mass of 2 mg of HA was inserted in a centrifuge tube, then dispersed in 1 mL of Milli-Q water (2 mg/mL) overnight with magnetic stirring at 25 °C. A mass of 10 mg of BPPB was easily inserted in a second weighted centrifuge tube, then a solution of 1 mg/mL was obtained by dissolving it in 10 mL of Milli-Q water. The HA solution was added to the BPPB solution obtaining a composite solution (1.09 mg/mL). The mixture corresponded to a 1/5 HA/BPPB wt. ratio, whose total initial weight (*Wi*) at the fully dry state was 12 mg. The mixture was sonicated for 30 min while heated at 60 °C, observing the formation of a gel (HA-BPPB-HG).

#### Reaction Work-Up and Recovery of HA-BPPB-HG

The gel was deprived by the not-entrapped water and was recovered at its maximum water content by centrifugation, while the BPPB possibly not entrapped in the formed 3D network of the achieved HA-BPPB-HG remained dissolved in the water which was removed just by the centrifugation. The final weight (*Wf*) of HA-BPPB-HG was therefore obtained upon the following operations. Centrifugation was conducted at 4000 rpm for 30 min, and for an additional 20 min at 4000 rpm. The not-absorbed water was removed and the tube containing the soaked HA-BPPB-HG was left upside down. The sample was weighted over time until a constant weight was reached (*Wf*). *Wi* and *Wf* were used to have an early idea of the *WAC* (%), calculable according to the following formula (Equation (5)) and later determined starting from a fully dried gel, adding an excess of water, and standing overnight (Section 2.4.5).
(5)$$WAC\;(\%)=\frac{Wf-Wi}{Wi}\times 100\;$$
where *Wi* and *Wf* are the volume of the dried HA-BPPB-HG and that of the swollen material, respectively. The reaction was repeated five times to evaluate the reproducibility of the results. Swollen HA-BPPB-HG samples from all the reactions were fully dried by freeze-drying and the highly hygroscopic dried hydrogels were stored separately in a dryer under vacuum for other experiments. Once characterization concluded, they were collected using water and fully dried by freeze-drying again, obtaining about 55 mg of fully dried hydrogel.

### 3.4. Characterization of HA-BPPB-HG

#### 3.4.1. ATR-FTIR of HA, BPPB and HA-BPPB-HG

ATR-FTIR analyses were carried out on HA, BPPB, and on fully dried HA-BPPB-HG, directly on the solid samples. The spectra were acquired from 4000 to 600 cm^−1^, with a 1 cm^−1^ spectral resolution, co-adding 32 interferograms, and with a measurement accuracy in the frequency data at each measured point of 0.01 cm^−1^ due to the internal laser reference of the instrument. Acquisitions were made in triplicate, and the spectra shown in the Discussion Section are the most representative images. Subsequently, we arranged the ATR-FTIR data of the spectra acquired for all the samples in a matrix of 3401 × 3 of 10,203 measurable variables. The obtained matrix was processed by Principal Components Analysis (PCA), using CAT statistical software (Chemometric Agile Tool, freely downloadable online at: http://www.gruppochemiometria.it/index.php/software/19-download-the-r-based-chemometric-software; current version 8 September 2024).

#### 3.4.2. Assessment of BPPB Content in HA-BPPB-HG

First, the BPPB calibration curve was constructed. A stock solution of BPPB (2 mg/mL; 2.3 µM) was prepared dissolving BPPB (10 mg) in MeOH (5 mL). Then, upon proper dilutions with MeOH, standard solutions at BPPB concentrations of 0.5, 0.25, 0.125, 0.0625, 0.0313, and 0.0156 mg/mL were obtained. The BPPB solutions were analyzed using a Fiber Optic UV-Vis Spectrometer System Ocean Optics USB 2000 (Ocean Optics, Inc., Dunedin, FL, USA) in 3 mL quartz cuvettes, detecting the related absorbance (Abs) at room temperature and at 267 nm. A solution of MeOH not containing BPPB was used as blank. Determinations were made in triplicate, and results were reported as mean of three independent experiments ± SD. The BPPB concentrations were plotted vs. the Abs values, and the BPPB calibration curve (Appendix A) was obtained by least-squares linear regression analysis using Microsoft Excel 365 software. Equation (6) of the developed linear calibration model was the following.
(6)y=1.7363×x+0.4306 

In Equation (6), *y* is the Abs value measured at 267 nm and *x* is the BPPB standard concentration analyzed. In Equation (6), the slope represents the coefficient of extinction (ε = 1.7363) of BPPB.

##### Estimation of BPPB Entrapped in the 3D Network of HA-BPPB-HG

To estimate the amount of BPPB contained in HA-BPPB-HG, 1.2 mg of fully dried gel, were dispersed in 5 mL of MeOH, obtaining a solution 0.24 mg/mL, and vigorously stirred for ten minutes to promote the release of BPPB. The clear solution was divided in 5 aliquots of 1 mL each, which were then subjected to UV-Vis analysis. The amount of BPPB in the aliquots was quantified at 267 nm. Determinations were consequently repeated five times, and results were reported as mean of five independent experiments ± SD.

##### Drug Loading (DL%) and Encapsulation Efficiency of HA-BPPB-HG

The drug loading (*DL%*) and encapsulation efficiency (*EE%*) values of HA-BPPB-HG were calculated from the following (Equations (7) and (8)) [50].
(7)$$DL\;\left(\%\right)=\frac{Weight\;of\;BPPB\;in\;the\;Dried\;Gel}{Weight\;of\;Dried\;Gel}\times 100$$
(8)$$ML\;\left(\%\right)=\frac{Weight\;of\;BPPB\;in\;the\;Total\;Dried\;Gel}{Weight\;of\;Loaded\;BPPB}\times 100$$

##### Statistical Analysis

The statistical significance of the slope of the BPPB-calibration curve was investigated through the analysis of variance (ANOVA) and performing the Fischer test. Statistical significance was established at the *p*-value < 0.05.

#### 3.4.3. Scanning Electron Microscopy (SEM)

The microstructure of the lyophilized hydrogel (HA-BPPB-HG) was investigated by SEM analysis. The experiments were carried out as previously reported [93]. Briefly, the fully dried hydrogel samples were fixed on aluminum pin stubs and sputter-coated with a gold layer of 30 mA for 1 min to improve the conductivity, and an accelerating voltage of 20 kV was used for the sample’s examination using a ZEISS SUPRA 40 VP field emission scanning electron microscope (Zeizz s.p.a., Milan, Italy). The micrographs were recorded digitally using a DISS 5 digital image acquisition system (Point Electronic GmbH, Halle, Germany). Images were acquired at 175, 250, 274, and 1000× magnification. Magnifications greater than 6000× caused the collapse of the hydrogel 3D network, due to the highly focused electron beam. As a result, examinations at higher magnifications were not performed.

#### 3.4.4. Equilibrium Swelling Rate

The swelling measurements were carried out at room temperature by immersing 10.0 mg of fully dried HA-BPPB-HG in deionized water (pH = 4 and 10) or in PBS (pH = 7.4) in a test tube [82]. At intervals of time selected according to the literature, weight measurements were made until the weight was constant [81]. The cumulative swelling ratio percentage (*Q* %), as a function of time, was calculated from (Equation (9)).
(9)$$Q\;\left(\%\right)t=\frac{Ws(t)-Wd}{Wd}\times 100$$where *Wd* and *Ws*(*t*) are the weights of the fully dried and swollen hydrogel at time *t*, respectively. The equilibrium swelling rate (*Q* (%) *e*) and the equilibrium degree of swelling (*EDS* (%)) was determined at the time point (time *t*) the hydrated HA-BPPB-HG samples achieved a constant weight.

#### 3.4.5. Water Absorbing Capacity (WAC (%))

For determining the water absorbing capacity (WAC (%)) of HA-BPPB-HG, samples (1–2 mg) of fully dried gel were placed in a graduated centrifuge tube (Ø ext. = 14 mm) and exactly weighed (*Wi*). An excess of water was added (10–12 mL) and left at room temperature overnight, under gentle magnetic stirring. Upon centrifugation at 4000 rpm for 20 min, and for an additional 20 min at 4000 rpm, the not-absorbed water was removed, and soaked samples of HA-BPPB-HG were left upside down. The samples were weighted over time until a constant weight was reached (*Wf*). The values of *Wi* and *Wf* were used to calculate WAC (%) as reported in Section 2.2.

#### 3.4.6. Porosity

To determine the porosity (P) of HA-BPPB-HG, intended as its equilibrium water content (EWC), the method described by Liu et al. was used [94]. We proceeded as in the experiment previously described, and *EWC* (%) and *P* (%) were calculated from the following (Equations (10) and (11)):(10)$$EWC\;\left(\%\right)=\frac{W2-W1}{W2}\times 100$$
(11)$$P\;\left(\%\right)=\frac{W2-W1)}{\rho V2}\times 100$$
where *W*1 is the weight of HA-BPPB-HG before its immersion in water, *W*2 is the weight of the hydrogel after its immersion in water overnight, *ρ* is density of water (*ρ* = 1), and *V*2 is the volume of the soaked gel after standing in water overnight.

#### 3.4.7. Biodegradability of HA-BPPB-HG over Time by In Vitro Mass Loss Experiments

The biodegradability in water over time of samples of HA-BPPB-HG was assessed in vitro at pH = 4, 7, and 10. Briefly, samples of vacuum-dried HA-BPPB-HG were inserted in centrifuge tubes, added with 10 mL of water solutions (pH = 4 or 10) or PBS (pH = 7.4), and incubated at 37 °C. At fixed time points of 6, 12, 24, 48, 72, 96, 144, and 192 h (8 days), the hydrogels formed in the tubes were centrifugated to remove the supernatant liquid, and the water residual on the hydrogels surface was wiped off by inverting the tubes upside down. The hydrogels were then dried and weighed to record their mass change (mass loss) over time. The weight of the hydrogels at the pre-set time intervals has been indicated as *Mt*. The cumulative mass loss percentages (*ML* (%)) over time were calculated according to the following (Equation (12)):(12)$$ML\;\left(\%\right)=\frac{Mi-Mt}{Mi}\times 100$$

In (10), *Mi* and *Mt* are the initial mass of the dry HA-BPPB-HG and the mass of remaining 3D structure after standing a time *t* in the aqueous medium, respectively.

#### 3.4.8. Evaluation of in Vitro Releases of BPPB

The in vitro release of BPPB from HA-BPPB-HG was determined by weighting 4.2 mg of fully dried HA-BPPB-HG, whose determined drug loading was 82.7% (83%), i.e., an assumed BPPB amount of about 3.5 mg. The sample was immersed in 10 mL of PBS medium (pH = 7.4) in a dialysis bag with 3.5 K MWCO (Cellu Sep H1, Orange Scientific, Braine-l’Alleud, Belgium) and subsequently dialyzed against 30 mL of release medium (pH = 4 and 7 (PBS)) at 37 °C and 100 rpm. The basic pH tested in the degradation experiments was not further considered because it is a condition unlike to occur in the tumor environment. At fixed interval points (0, 1, 2, 4, 24, 28, 48, 72, and 96 h), aliquots of 2 mL of the release medium were taken out and 2 mL of fresh PBS or acidic solution were replenished. The amount of BPPB in the aliquots was detected by measuring their absorbance at 267 nm using the UV-Vis spectrophotometer previously described, while PBS or acidic solutions not containing BPPB were used as blanks. Determinations were made in triplicate, and results were expressed as mean of three independent experiments ± SD. The obtained concentrations were used to compute the cumulative drug release percentage, according to (Equation (13)):(13)$$CDR\;\left(\%\right)=\frac{Dt}{Di}\times 100$$

In Equation (13), *Dt* is the concentration of BPPB in the aliquots at incubation time *t*, while *Di* is the BPPB concentration in the dialysis tube, given by the total BPPB entrapped in the weight of HA-BPPB-HG analyzed according to the computed DL% when dissolved in 10 mL of PBS (0.3486 mg/mL).

## 4. Conclusions

In this study, a new potent anticancer compound with concentration- and mainly time-dependent antiproliferative effects on two cutaneous metastatic melanoma (CMM) BRAF mutant cell lines, has been identified in a TPP-based bola amphiphile (BPPB), already reported for its antibacterial effects against MDR clinical isolates of ESKAPE bacteria. Even if its powerful antitumor equipment was previously detected by us on sensitive and MDR neuroblastoma (NB) cells, assessing very low IC_50_ values of 0.2 µM on HTLA-230 regardless the exposure timing, even more low IC_50_ values of 0.049 µM were in this case determined on MeOV CMM cell line after 72 h exposure. Differently from results on NB, the effects of BPPB on CMM cells were strongly dependent on exposure timing and the IC_50_ values decreased by 6.9–17.9 times, depending on cell line, passing from 24 to 72 h of exposure. Despite this scenario on CMM cells, the selectivity indexes for these cells in relation to BPPB cytotoxicity vs. non-tumoral and non-immortalized mammalian cells (MRC-5) and RBCs were significantly augmented (2.3–5.7 and 6.9–16.9 times, depending on the types of CMM and non-tumoral type of cells considered), passing from 24 to 72 h of exposure. Excited from these findings and looking forward to a future dermal use of BPPB to treat and reduce the possible pigmentations caused by BRAF mutant CM and CMM cells, which usually appear on the skin of the head, abdomen, legs, and under the armpits, BPPB was formulated as a hydrogel using only HA as the jelling agent. A dosage form of BPPB was obtained with all requirements, such as high drug loading, a protracted release of BPPB, favorable biodegradability over time, spreadable behavior, high swelling capability, and high porosity, suitable for topical application. It is the third time that the very powerful biological effects of BPPB, a TPP-based bola amphiphilic molecule, have been unequivocally demonstrated, thus evidencing the very appealing potentialities of these type of compounds as per se active biomedical devices, only recently investigated. We hope that these three works and the next one on the specific molecular mechanism of action of BPPB could stimulate researchers to more extensively study such molecules.

## Data Availability

Data supporting reported results can be found in this article and in the related Appendix A.

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
