# Peer review of "Strongly ROS-Correlated, Time-Dependent, and Selective Antiproliferative Effects of Synthesized Nano Vesicles on BRAF Mutant Melanoma Cells and Their Hyaluronic Acid-Based Hydrogel Formulation"

_ijms, 2024, doi:10.3390/ijms251810071_

Round 1
Reviewer 1 Report
Comments and Suggestions for Authors
It was a pleasure to review the article submitted by Alfei et al. I believe that after a minor review it should be considered for publication in IJMS.
Minor:
1) There is an inconsistency in space separation before references, please correct
2) Figure 3, 4, 8 – x-legend is incorrect, please modify
3) Figure 5 - y-legend is incorrect, please modify
4) Axis captions should be unified, unites are sometimes in squared brackets and sometimes in rounded brackets
5) Were those melanoma cells exhibiting pigmentation? If so, this should be commented in the article.
6) Bright-field or fluorescent images of cells exposed to BPPB and control cells should be demonstrated in supplementary materials
7) "Acknowledgements" is highlited in yellow, please correct
Author Response
It was a pleasure to review the article submitted by Alfei et al. I believe that after a minor review it should be considered for publication in IJMS.
The Authors thank a lot the Reviewer for her/his positive comments on our study and hope the changes made it upon the other suggestions could satisfy him/her.
Minor:
1) There is an inconsistency in space separation before references, please correct.
The Reviewer is right, and we have tried to normalize the inconsistency in space separation before references, even if to adjust such problems, it will be a duty of the Editorial Office during proofs preparation.
2) Figure 3, 4, 8 – x-legend is incorrect, please modify
The legends on x axis in the signaled Figures 3, 4 and 8 seem correct. The errors were in the captions of Figure 3 and 4, where “PBBP” was used in place of “BPPB”. The issue has been corrected. In Figure 8, both x axis legend and its caption seem correct, and we did not find any error. In fact, they are identical to those of Figure 7, that the Reviewer has retained correct. Anyway, we have added further specifications to the captions of Figures 3, 4, 7 and 8, to better address the Reviewer comment.
3) Figure 5 - y-legend is incorrect, please modify
In Figure 5 y axis legend, i.e. IC50 (µM), we did not find any error. Anyway, we have added further specifications to better address the Reviewer comment.
4) Axis captions should be unified, unites are sometimes in squared brackets and sometimes in rounded brackets
The Reviewer comment refers to Figure 11, which is the only one providing the unit of the measures in squared brackets, due to the not changeable defaults set up of the plot appearance, imposed by the software used to elaborate the ATR-FTIR spectra (SpectraGryph 1.2, promo edition).
In all other cases, the measure units are in rounded brackets. We kindly ask the Reviewer to not force us to use another software to satisfy him/her.
5) Were those melanoma cells exhibiting pigmentation? If so, this should be commented in the article.
A special thank to the Reviewer for his/her suggestion. The required information has enriched our study. Please, consider lines 213-214, 237-239 and 1059-1061.
6) Bright-field or fluorescent images of cells exposed to BPPB, and control cells should be demonstrated in supplementary materials
We thank the Reviewer for his/her suggestion which could further enrich this study, and we are very happy to find that the Reviewer is of the same opinion of us, concerning the additional experiments which are needed to be carried out with BPPB, before passing to experiments on MDR CMM cells, we have available only now, and to animal models. Nevertheless, fluorescent images were not in the scope of the present work. We reveal to the Reviewer, that BPPB was already successfully marked at > 95% with the green, fluorescent lipophilic dye (namely Dio), to be exposed to red marked CMM cells and to confirm BPPB accumulation in mitochondria. Unfortunately, the product useful to mark CMM cells will available only next week, due to the closure of the manufacturer for holydays. Results from these experiments and other exciting ones will be provided in our next work, where the complete mechanism of action of BPPB will be reported, explained and discussed.
7) "Acknowledgements" is highlited in yellow, please correct
We apologize to the Reviewer for our distraction. The yellow highlighting has been removed.
Reviewer 2 Report
Comments and Suggestions for Authors
The work by Alfei et al. concerns strongly ROS-correlated, time-dependent and selective anti-proliferative effects of synthesized nano vesicles on BRAF mutant melanoma cells and their hyaluronic acid-based hydrogel formulation. However, some changes have to be entered into the revised version of the manuscript before it can be further processed:
1. Figure 11 - increase the y-axis because the peaks are too small and there is no difference in the spectra
2. present all equations in equation mode, not sequence mode
3. line 281 – “PBBP” – is it an editorial error or new substance?
4. the authors presented the activity of Cytotoxic Effects of BPPB on cells; the same studies are required for the formulation and the formulation base - the negative influence of the base components must be excluded
Author Response
The work by Alfei et al. concerns strongly ROS-correlated, time-dependent and selective anti-proliferative effects of synthesized nano vesicles on BRAF mutant melanoma cells and their hyaluronic acid-based hydrogel formulation. However, some changes have to be entered into the revised version of the manuscript before it can be further processed:
- Figure 11 - increase the y-axis because the peaks are too small and there is no difference in the spectra
We thank a lot the Reviewer for his/her witted observation. Anyway, to make more evident the difference in the bands of HA, HA-BPPB-HG and BPPB and to not burden further the main text with more than one image of ATR-FTIR spectra, additional images, where only couples of spectra in place of three spectra are visible, had been already provided in the Supplementary Materials file as Figure S8 and S9, in the original submission. To increase the y-axis without deforming the image is impossible.
- present all equations in equation mode, not sequence mode
The Reviewer is right, sorry for this inconsistence. The Reviewer request has been addressed.
- line 281 – “PBBP” – is it an editorial error or new substance?
It is an Editorial error, as other ones in the main text, which have been introduced erroneously by the Editorial Office, in meantime you revised. All errors have been corrected.
- the authors presented the activity of Cytotoxic Effects of BPPB on cells; the same studies are required for the formulation and the formulation base - the negative influence of the base components must be excluded
The Reviewer request is in part acceptable, even if premature for this study. We kindly explain to the Reviewer, that experiments using a hydrogel formulation are not possible and/or are not scientifically reliable if carried out directly in vitro on cells, as done for BPPB. For definition, gels are insoluble 3D networks and are not administrable dissolved in the culture medium of cells. Synthetic models of human skin and scratches tests are needed to test the possible cytotoxic or even proliferative effects of a hydrogel preparation (https://doi.org/10.1111/srt.12235, 10.1055/s-0028-1084679), as well as those of BPPB gel formulation developed here, but such advanced experiments were not in the scopes of the present work. Here, we have found an active principle, with very potent effects and high selectivity against two BRAF mutant CMM cells, due to its low cytotoxicity vs. even four mammalians not tumoral cell lines. We have confirmed its ROS-dependent mechanism of action by proper experiments. Finally, we have formulated BPPB as hydrogel, which was also fully characterized. Do the Reviewer still think that in this study could be place also for experiments using HA-BPPB-HG on artificial skin? Additionally, concerning the base components of the HA-BPPB-HG, i.e. hyaluronic acid (HA) only, have the Reviewer still doubts about its cytocompatibility being it, one of the constituents of the extracellular matrix (ECM)? In sight of how many articles are present in literature reporting on such question, we think that performing experiments to further assess HA low cytotoxicity could be redundant and boring. Please, consider only as few examples https://doi.org/10.3390/ma16145189, https://doi.org/10.1016/j.carbpol.2010.06.042, https://doi.org/10.1016/j.carbpol.2022.120153.
Round 2
Reviewer 2 Report
Comments and Suggestions for Authors
Accept in present form